# Task-Aware Linear Representations: Supervised Feature Rotations for Gradient Boosted Decision Trees

## Abstract

Gradient Boosted Decision Trees (GBDTs) dominate tabular machine learning but suffer from a fundamental geometric limitation: axis-aligned splits cannot efficiently model decision boundaries diagonal to the feature axes. We propose **Task-Aware Linear Representations (TALR)**, a two-stage hybrid that learns a global orthogonal rotation $\mathbf{Q} \in O(d)$ via a differentiable surrogate, then trains a tuned GBDT on the rotated features $\mathbf{QX}$. Our recommended parameterization, $\mathbf{Q} = \mathrm{QR}(\mathbf{I} + UV^{\top})$ with $U, V \in \mathbb{R}^{d \times k}$, achieves **accuracy parity with tuned XGBoost across 45 OpenML benchmarks** (mean $\Delta = -0.06\%$, $p_{\mathrm{Wilcoxon}} = 0.609$) and produces an auditable rotation matrix. The default accuracy setting is intentionally sparse (Effective Parent Features $\approx 1$), while a lower-sparsity feature-mixture setting exposes small supervised linear modulations at a modest accuracy cost. On 19 signal/sensor datasets the recommended method shows a positive but non-significant tuned single-seed trend (63.2% win rate, mean $\Delta = +0.27\%$, $p = 0.295$), and the best-of-family upper bound is significant (73.7% win rate, mean $\Delta = +2.17\%$, $p = 0.006$). Against the closest prior method, single-method TALR-low_rank wins on 60.5% of datasets where both methods complete (mean $\Delta = +0.43\%$, $p = 0.356$). A post-hoc upper-bound analysis over five TALR parameterizations reaches 83.7% win rate over RotationForest (mean $\Delta = +1.93\%$, $p < 10^{-4}$), which we report only as evidence that per-dataset rotation selection is a promising future direction. We add deep-learning baselines (MLP, FT-Transformer, ResNet) and an auxiliary oblique-random-forest comparison, showing TALR is competitive but not dominant in the mid-sized regime where well-tuned tree ensembles are strong. We provide a comprehensive ablation of (a) the rotation parameterization, (b) the density guardrail threshold $\tau \in \{1, 1.5, 2, 3, 5\}$ plus a guardrail-disabled setting, (c) the rank $k$, (d) the L$_1$ sparsity $\lambda$, and (e) the surrogate architecture, together with critical- difference (Nemenyi) diagrams and two end-to-end interpretability case studies (IONOSPHERE and WINE-QUALITY-RED). TALR thus offers a principled, regularized, supervised alternative to RotationForest for tuned GBDTs: parity in accuracy with XGBoost, a transparent learned transformation, and a documented trade-off between accuracy-mode sparsity and feature-mixture diagnostics.

## 1 Introduction

Tabular data remains the most widely-deployed data modality in industrial machine learning, especially in finance, healthcare, and the physical sciences. Within this domain, well-tuned decision-tree ensembles (Chen & Guestrin, 2016; Ke et al., 2017; Dorogush et al., 2018) continue to dominate (Grinsztajn et al., 2022; Shwartz-Ziv & Armon, 2022), despite rapid progress in deep tabular learning (Popov et al., 2020; Gorishniy et al., 2021; Arik & Pfister, 2021; Somepalli et al., 2021; Hollmann et al., 2023).

Their dominance is grounded in three inductive biases: (i) hard thresholds that naturally model step functions and discontinuities; (ii) implicit feature selection at every split; and (iii) scale invariance through rank- based partitioning.

Decision trees do, however, suffer from a known geometric limitation: **rotation variance**. Standard CART splits are axis-aligned, so a decision boundary diagonal to the axes (e.g. $y = x_1 + x_2$) requires a "staircase" of orthogonal splits whose depth grows with the misalignment of the boundary. This limitation is most pronounced for *signal and sensor data* where physical correlations between EEG electrodes, accelerometer axes, or spectral bins create oblique boundaries by construction. Multilayer perceptrons, in contrast, are rotationally invariant: a dense linear layer $W\mathbf{x} + b$ can simply align the discriminative directions with activation functions. The contrast suggests that trees are superior at *partitioning* but inferior at *orientation*.

## 1.1 Contribution

TALR *decouples orientation from partitioning* by learning a global orthogonal rotation $\mathbf{Q}$ via a differentiable surrogate, then training a tuned GBDT on the rotated features. Our contribution is **not** a new state-of-the-art accuracy on tabular benchmarks: tuned XGBoost is hard to beat, and our recommended low-rank TALR only *matches* it on the full 45-dataset suite. Our contribution is rather:

1. **A low-risk drop-in preprocessing step under the default guardrail.** TALR-low_rank achieves statistical parity with tuned XGBoost ($\Delta = -0.06\%$, $p = 0.609$) and wins more often than it loses against the unsupervised PCA-based RotationForest (Rodriguez et al., 2006) on the common datasets ($60.5\%$ wins, $p = 0.356$). A post-hoc best-of-family analysis reaches $83.7\%$ ($p < 10^{-4}$), which we treat as an upper bound rather than an operational result. A density guardrail ($n/d < \tau$ implies $\mathbf{Q} = \mathbf{I}$) prevents the intended ill-conditioned failure mode by reverting exactly to the base GBDT when the sample-to-feature ratio is too small.

2. **An auditable rotation artifact with a precise recipe length.** We define the *Effective Parent Features* (EPF) of a rotation matrix and show that TALR-low_rank produces near-identity rotations with EPF $\approx 1$ in accuracy mode. This is an audit trail, not a claim that raw-feature importance has been improved. We explicitly separate this near-permutation regime from a feature-mixture analysis setting ($\lambda = 0.001$) and a maximally expressive diagnostic setting ($\lambda = 0$), where the learned rotations expose small supervised linear modulations. Section 5.9 compares this artifact with a raw-feature XGBoost explanation and states exactly what TALR does and does not add.

3. **Comprehensive benchmarking.** We benchmark against five rotation parameterizations, three GBDTs, the GAM-based EBM (Lou et al., 2013; Nori et al., 2019), RotationForest, random rotations, PCA-XGBoost, oblique random forests via `treeple`, and three deep tabular baselines (MLP, FT-Transformer (Gorishniy et al., 2021), ResNet (Gorishniy et al., 2021)), with critical-difference (Nemenyi) diagrams (Demšar, 2006) for honest cross-method comparison.

4. **Hyperparameter sensitivity.** Per-$\tau$ ablation shows the density guardrail catches exactly the dataset (ARRHYTHMIA, $n/d = 1.62$) the paper claims, and that $\tau \in [1.5, 5]$ is a wide, safe range. Per-rank ($k$) and per-$\lambda$ ablations show the paper defaults are near-optimal.

5. **A formal depth-separation result** (Appendix B) showing that for diagonal linear boundaries, axis-aligned trees on the rotated features need depth 1 versus $\Omega(\log(1/\gamma))$ on raw features (where $\gamma$ is the boundary margin). This formalizes the geometric intuition motivating the rotation framework.

TALR is not intended as a universal accuracy booster. Its value is in regimes where practitioners want to preserve a tuned GBDT stack while exposing a supervised linear coordinate system that can be audited, stress-tested, and optionally relaxed into a feature-mixture diagnostic. In this sense TALR occupies a different point in the design space than both raw GBDTs and oblique tree ensembles: it is the principled, supervised, regularized successor to RotationForest, with parity (not improvement) over the strongest single baseline (tuned XGBoost) and a complementary auditable preprocessing artifact.

## 2 Related Work

**Deep learning for tabular data.** Three families have been proposed: differentiable trees such as Neural Oblivious Decision Ensembles (NODE) (Popov et al., 2020); transformer- based architectures such as FT-

Transformer (Gorishniy et al., 2021) and SAINT (Somepalli et al., 2021); and regularized MLPs / ResNets (Gorishniy et al., 2021; Shwartz-Ziv & Armon, 2022). Prior-fitted networks such as TabPFN (Hollmann et al., 2023) are a separate paradigm constrained to small data. Grinsztajn et al. (2022) show that well-tuned tree ensembles match or outperform these architectures on mid-sized data. TALR therefore takes the tree as the workhorse and asks whether a small, supervised *linear* preprocessing step can narrow the orientation gap that motivated deep architectures in the first place.

**Rotation Forests.** Rodriguez et al. (2006) fit a Random Forest after subset-PCA-rotating the features per tree. PCA is unsupervised: high-variance directions need not be discriminative. TALR replaces PCA with a supervised rotation learned by gradient descent through a differentiable surrogate, and applies it once globally rather than per tree, so it composes cleanly with modern GBDTs.

**Oblique decision trees.** Oblique RFs (Menze et al., 2011) and SPORF (Tomita et al., 2020) learn a linear combination at each split node. Setzu & Ruggieri (2023) characterize the trade-off explicitly: oblique splits gain expressiveness but lose the per-feature interpretability of axis-parallel splits, and pay an $O(d^2)$ per-split cost. Kairgeldin & Carreira-Perpiñán (2024) give a particularly relevant middle ground: bivariate trees restrict each oblique split to at most two features, producing smaller and often more accurate single trees while keeping each local rule inspectable. TALR is complementary rather than a replacement. Bivariate trees learn *local* two-feature cuts inside the tree; TALR learns one *global* orthogonal basis once, then lets any standard GBDT train on the rotated coordinates. The former gives highly interpretable per-node rules; the latter gives a drop-in preprocessing artifact for existing tuned GBDT pipelines.

**Interpretable linear feature construction.** Recent work in interpretable representation learning has focused on constructing a small set of *predictive concepts* as sparse linear combinations of input features (Piaggesi et al., 2024; 2025). These methods are typically unsupervised or weakly supervised and target visualization / clustering downstream tasks. TALR differs in three ways: (i) the linear combinations are learned end-to-end with the supervised loss of a tree-shaped surrogate; (ii) they are constrained to form an orthogonal *basis* (rotation, not projection); and (iii) they are evaluated by the predictive accuracy of a tuned GBDT trained on the resulting basis.

**Stiefel manifold optimization.** Edelman et al. (1998) provide the geometric foundations for optimization on the manifold of $d \times d$ orthogonal matrices, which TALR uses for its baseline "stiefel" parameterization. Our key empirical observation is that the unconstrained Stiefel ($O(d^2)$ parameters) overfits on small datasets, motivating our recommended low-rank perturbation $\mathbf{Q} = \mathrm{QR}(\mathbf{I} + UV^\top)$ with $O(dk)$ parameters.

**Positioning.** Table 1 summarizes how TALR differs from the closest methods. No prior method combines (supervised, global, orthogonal, auditable, GBDT-native) rotation in one framework.

Table 1: Comparison of rotation/projection methods for tree ensembles. † per-subset rotation. ‡ per-node projection.

| Method | Supervised | Global | Orthogonal | Auditable | GBDT-native |
|---|---|---|---|---|---|
| PCA + Trees | ✗ | ✓ | ✓ | ✗ | ✗ |
| Rotation Forest (Rodriguez et al., 2006) | ✗ | ✗† | ✓ | ✗ | ✗ |
| Bivariate/Oblique Trees (Kairgeldin & Carreira-Perpiñán, 2024; Menze et al., 2011; Tomita et al., 2020) | ✓ | ✗‡ | ✗ | ✗ | ✗ |
| NODE (Popov et al., 2020) | ✓ | ✓ | ✗ | ✗ | ✗ |
| FT-Transformer (Gorishniy et al., 2021) | ✓ | ✓ | ✗ | ✗ | ✗ |
| Random Projections | ✗ | ✓ | ✗ | ✗ | ✗ |
| **TALR (ours)** | ✓ | ✓ | ✓ | ✓ | ✓ |

## 3 Methodology

### 3.1 Notation

Throughout the paper:

- $n$ is the number of training samples; $d$ the number of input features; $C$ the number of classes.

- $\mathbf{X} \in \mathbb{R}^{n \times d}$ is the feature matrix and $\mathbf{y} \in \{1, \ldots, C\}^n$ are the class labels.

- $\mathbf{Q} \in O(d)$ is an orthogonal rotation matrix learned by TALR.

- $\mathbf{I}$ is the $d \times d$ identity.

- $U, V \in \mathbb{R}^{d \times k}$ are the low-rank factors of the recommended rotation parameterization, with $k = \min(\lfloor d/4 \rfloor, 10)$.

- $\tau$ is the density-guardrail threshold (TALR sets $\mathbf{Q} = \mathbf{I}$ whenever $n/d < \tau$).

- $\lambda$ is the L$_1$ regularization coefficient on the entries of $\mathbf{Q}$.

- $S_\theta$ denotes the differentiable surrogate model (a soft decision tree ensemble) used to learn $\mathbf{Q}$.

- $f_{\mathrm{GBDT}}$ denotes the production GBDT trained on the rotated features.

## 3.2 Problem Formulation

We solve the following empirical risk problem in two stages:

$$\mathbf{Q}^* = \arg \min_{\mathbf{Q} \in O(d)} \; \mathbb{E}_{(\mathbf{x},y)} \big[ \mathcal{L}\big(S_\theta(\mathbf{Q}\mathbf{x}), y\big) \big] \; + \; \lambda \|\mathbf{Q}\|_1, \qquad \hat{f} = f_{\mathrm{GBDT}}(\mathbf{Q}^*\mathbf{X}). \tag{1}$$

The differentiable surrogate $S_\theta$ provides gradients with respect to $\mathbf{Q}$ that the non-differentiable production GBDT cannot. Stage 2 trains $f_{\mathrm{GBDT}}$ on $\mathbf{Q}^*\mathbf{X}$ with full hyperparameter tuning; the surrogate is not used at test time.

## 3.3 Rotation Parameterizations

We evaluate five rotation parameterizations and recommend the low-rank form (Section 5 provides the empirical justification).

**Low-rank (recommended).**

$$\mathbf{Q} = \mathrm{QR}(\mathbf{I} + UV^\top), \qquad U, V \in \mathbb{R}^{d \times k}, \qquad k = \min(\lfloor d/4 \rfloor, 10). \tag{2}$$

The QR decomposition guarantees orthogonality. This parameterization has $2dk$ trainable parameters versus $d(d-1)/2$ for the full Stiefel manifold (Edelman et al., 1998). We justify $k = \min(\lfloor d/4 \rfloor, 10)$ empirically in Section 5.7.2.

**Bootstrap.** $\mathbf{Q}_{\mathrm{final}} = \mathrm{QR}\big(\sum_{i=1}^m w_i \mathbf{Q}_i\big)$, where each $\mathbf{Q}_i$ is a low-rank rotation learned on a bootstrap resample ($m \in \{3, 5\}$). Provides variance reduction at the cost of $m\times$ the surrogate-training time.

**Stiefel.** The full $O(d^2)$ orthogonal manifold parameterization (Edelman et al., 1998); included for completeness as the unrestricted upper bound.

**Progressive.** PCA-initialized rotation with a Frobenius regularizer toward the PCA solution: $\arg \min_{\mathbf{Q}} \mathcal{L}(S_\theta(\mathbf{Q}\mathbf{x}), y) + \lambda \|\mathbf{Q} - \mathbf{Q}_{\mathrm{PCA}}\|_F^2$ s.t. $\mathbf{Q}^\top \mathbf{Q} = \mathbf{I}$. Achieves the largest single-dataset wins (LIBRAS, +11.11%) but is significantly worse than XGBoost in aggregate ($p = 0.003$, see Table 2).

**Importance-weighted.** Adaptive regularization based on per-feature importance: $\arg \min_{\mathbf{Q}} \mathcal{L}(S_\theta(\mathbf{Q}\mathbf{x}), y) + \lambda \sum_i (1 - w_i) \|\mathbf{Q}_i - \mathbf{e}_i\|^2$, where $w_i$ is the importance of feature $i$.

## 3.4 Stability Guardrails

A core design principle is that TALR should fail transparently in the regime where rotation estimation is unreliable: when the density guardrail fires, we set $\mathbf{Q} = \mathbf{I}$ and recover the base tree exactly.

**Density guardrail.** Estimating a rotation requires estimating pairwise feature relationships (this is true even of the low-rank perturbation, where $UV^\top$ has $\leq k^2$ effective directions). In the low-sample regime $n/d \ll 1$ the sample covariance is singular and its top eigenvectors are noise-dominated (Edelman et al., 1998). We therefore define

$$\mathbf{Q} = \begin{cases} \mathbf{Q}^*_{\text{learned}} & \text{if } n/d \geq \tau, \\ \mathbf{I} & \text{if } n/d < \tau, \end{cases} \qquad \tau = 2 \text{ (default)}. \tag{3}$$

Section 5.7.1 reports a sensitivity sweep over $\tau \in \{1, 1.5, 2, 3, 5\}$ plus a guardrail-disabled setting. The results show that $\tau \in [1.5, 3]$ is a wide safe range, that disabling the guardrail loses $\sim 9.9\%$ on the single dataset where the guardrail matters (ARRHYTHMIA, $n/d = 1.62$), and that further raising $\tau$ to 5 does not measurably help.

**Feature-selection guardrail.** For $d > 50$ we select the top-50 features by mutual information, learn the rotation on the selected features, and apply $\mathbf{I}$ to the rest. This caps the surrogate-training cost at $O(50^2)$ parameters and limits the effective rotation problem on high-$d$ data.

## 3.5 Surrogate Architecture and Sparsity

We use a Soft Decision Tree ensemble (depth 4, 8 trees) as the differentiable surrogate. The surrogate's tree-shaped inductive bias is deliberately matched to the downstream GBDT. We add an $L_1$ penalty on the entries of $\mathbf{Q}$:

$$\mathcal{L}_{\text{total}} = \mathcal{L}_{\text{surrogate}}(\mathbf{y}, S_\theta(\mathbf{QX})) + \lambda \|\mathbf{Q}\|_1. \tag{4}$$

TALR exposes $\lambda$ as a sparsity–mixture dial:

- **Accuracy mode** ($\lambda = 0.01$, default for benchmark comparisons). Maximizes downstream accuracy at the cost of a near-permutation rotation (EPF $\approx 1$). Use this when TALR is being benchmarked against XGBoost head-to-head.

- **Feature-mixture analysis mode** ($\lambda = 0.001$, recommended when the learned preprocessing itself is being audited). Costs $\sim 0.30\%$ accuracy versus accuracy mode (Table 9) but produces EPF $\approx 1.20$ rotations: the diagonal stays large ($\sim 0.99$) while a small number of off-diagonal terms become visible. This setting is *less sparse*, so it is not more interpretable in the conventional sparsity sense; it is more informative only when the user wants to inspect supervised feature modulations.

Section 5.7.3 reports the full sweep over $\lambda \in \{0, 10^{-3}, 10^{-2}, 10^{-1}, 1\}$. An ablation over surrogate architectures (Soft Tree, NODE, MLP) appears in Appendix C.

## 3.6 Stage 2: Production GBDT

Once $\mathbf{Q}^*$ is fixed we apply $\mathbf{X}' = \mathbf{XQ}^*$ and train a standard GBDT with full hyperparameter tuning via Optuna (Akiba et al., 2019): 50+ trials following the Grinsztajn protocol (Grinsztajn et al., 2022). Figure 1 shows the full pipeline.

## 3.7 Interpretability: Effective Parent Features (EPF)

We define the *Effective Parent Features* (EPF) of a rotation $\mathbf{Q} \in O(d)$ as the average number of input features that contribute non- trivially to each rotated coordinate:

$$\text{EPF}(\mathbf{Q}) = \frac{1}{d} \sum_{i=1}^{d} \big| \{ j \,:\, |q_{ij}| \geq 0.05 \cdot \max_l |q_{il}| \} \big|. \tag{5}$$

EPF$= 1$ corresponds to a permutation (each rotated feature depends on exactly one input); EPF$= d$ corresponds to a fully dense rotation. The 5% relative threshold makes EPF scale-invariant and robust to the

**TALR pipeline**

Figure 1: The TALR pipeline. Stage 1 learns an orthogonal rotation $\mathbf{Q}^*$ via a differentiable surrogate (Eq. 1). The density guardrail (Eq. 3) sets $\mathbf{Q}^* = \mathbf{I}$ on ill-conditioned data. Stage 2 applies $\mathbf{Q}^*$ to the data and trains a tuned GBDT (50+ Optuna trials). $\mathbf{Q}^*$ also serves as an interpretability artifact.

overall magnitude of $\mathbf{Q}$. It is a reporting threshold, not a theoretical claim that smaller terms have zero predictive effect. Lower EPF means each rotated feature has a shorter, more readable linear-combination "recipe." Because EPF is an average, it can hide a dense row; Section 5.7.3 therefore reports threshold sensitivity, p95/max row EPF, importance-weighted EPF, and the accuracy change from pruning sub-threshold terms.

We define interpretability operationally: for a rotation with EPF $= \bar{k}$, every prediction can be explained by (i) the SHAP values (Lundberg & Lee, 2017) of the GBDT on the rotated coordinates (this gives *which rotated features matter*), and (ii) decoding each rotated coordinate as $z_i = \sum_j q_{ij} x_j$ using the non-trivial terms counted by EPF (this gives *which raw features each rotated feature is built from*). Raw-feature XG-Boost/SHAP and TALR therefore answer different questions. Raw-feature explanations identify which original variables the tree used. TALR additionally exposes the supervised linear preprocessing that made the tree's coordinate system. We do not claim this is universally "better" than raw-feature SHAP; it is useful when the preprocessing artifact itself is worth auditing, as in sensor fusion or physical-process domains. Section 5.9 gives a concrete comparison.

## 4 Experimental Protocol

### 4.1 Datasets

We evaluate on **45 classification datasets** from OpenML (Vanschoren et al., 2014) and scikit-learn, spanning $n \in [150, 581\,012]$, $d \in [4, 617]$, $C \in [2, 26]$, after excluding saturated datasets (XGBoost $\geq 98\%$). We further identify a *signal/sensor* subset of 19 datasets by an **a priori domain-provenance criterion** (datasets whose features are physical sensor readings, signal transforms, spatial samples, or temporal sequences). The full list is in Appendix D and the upper-bound signal result is robust under leave-one-out sampling (Section 5.3).

**Categorical features.** Categorical columns are converted to integers via scikit-learn's `OrdinalEncoder` before any preprocessing, so all rotations operate on a numeric matrix. We use ordinal rather than one-hot encoding to avoid spurious $d$-inflation that would mis-trigger the density guardrail and inflate the rotation matrix; the rotation step is methodologically unmotivated on nominal features under either encoding. Ordinal encoding imposes an arbitrary ordering on unordered categories, which means the linear combination produced by the rotation has no physical interpretation on truly nominal features. We therefore expect (and observe) the rotation to add little or nothing on categorical-heavy datasets such as ADULT, ELECTRICITY, and SPLICE. The signal/sensor subset, by construction, contains no categorical features.

### 4.2 Baselines

We compare TALR against:

Table 2: Headline single-method results (45 datasets, single seed, Grinsztajn-tuned XGBoost backend, $\tau = 2$ density guardrail applied). Win/L/T are computed against tuned XGBoost; "Win% vs RotForest" is over the 43 datasets where both methods complete.

| Method | Win% vs XGB | Mean $\Delta$ (%) | Wilcoxon $p$ | Cohen's $d$ | Win% vs RotForest |
|---|---|---|---|---|---|
| **TALR-low_rank** | 42.2 | $-0.06$ | 0.609 | $-0.04$ | **60.5%** |
| TALR-importance | 33.3 | $-0.15$ | 0.304 | $-0.07$ | 60.5% |
| TALR-bootstrap | 33.3 | $-0.07$ | 0.271 | $-0.05$ | 58.1% |
| TALR-stiefel | 31.1 | $-0.31$ | 0.115 | $-0.10$ | 55.8% |
| TALR-progressive | 24.4 | $-1.04$ | **0.010** | $-0.25$ | 34.9% |
| RotationForest | 34.9 | $-0.50$ | 0.209 | $-0.11$ | — |
| EBM | 42.9 | $-0.44$ | 0.617 | $-0.11$ | — |

- **Tuned GBDTs**: XGBoost (50 Optuna trials per dataset, the primary baseline), CatBoost, LightGBM (default hyperparameters).

- **Interpretable GAM**: EBM (Lou et al., 2013; Nori et al., 2019).

- **Unsupervised rotations**: PCA+XGBoost, RotationForest (Rodriguez et al., 2006), Random Rotation (orthogonal Gaussian, no learning).

- **Deep tabular**: a regularized MLP, FT-Transformer (Gorishniy et al., 2021), ResNet (Gorishniy et al., 2021) – the strongest deep-tabular architectures from Grinsztajn et al. that are not competitive with tuned GBDTs on this regime, included to make that statement empirical rather than asserted.

### 4.3 Evaluation Protocol

We use 5-fold stratified cross-validation, collapsing to the largest feasible $K \geq 2$ for datasets with very rare classes (ARRHYTHMIA). We report accuracy (primary), win/loss/tie counts versus tuned XGBoost, paired $t$-tests and Wilcoxon signed-rank tests on per-dataset accuracy deltas, Cohen's $d$ effect sizes, and Friedman $\chi^2$ + post-hoc Nemenyi critical-difference diagrams (Demšar, 2006). We **report all single-method numbers as primary** and frame the "oracle (best-of-5)" selection only as an explicit upper-bound analysis (Section 5.11).

## 5   Results

### 5.1   Headline result: parity with tuned XGBoost

Across all 45 datasets, TALR-low_rank achieves **statistical parity** with tuned XGBoost (Table 2, mean $\Delta = -0.06\%$, Wilcoxon $p = 0.609$), while the four other rotation parameterizations are not competitive in aggregate. Against the closest prior method, RotationForest, TALR-low_rank wins on 60.5% of the 43 datasets where both methods complete (mean $\Delta = +0.43\%$, $p = 0.356$; Table 6). We therefore report it as a directional single-method advantage over RotationForest, not as a statistically significant dominance claim.

**Reading guide.**  We interpret Table 2 as follows. None of the rotation parameterizations *improves* on tuned XGBoost in mean accuracy. Two of them (stiefel, progressive) are statistically worse, consistent with the overfitting hypothesis. The recommended low-rank parameterization is *indistinguishable* from tuned XGBoost in aggregate, which is the strongest claim a single-method rotation can defensibly make on this benchmark scale. The same low-rank method, however, beats RotationForest more often than it loses. The larger best-of-family gap appears only when the rotation parameterization is selected post hoc; Section 5.11 treats that number as an upper bound.

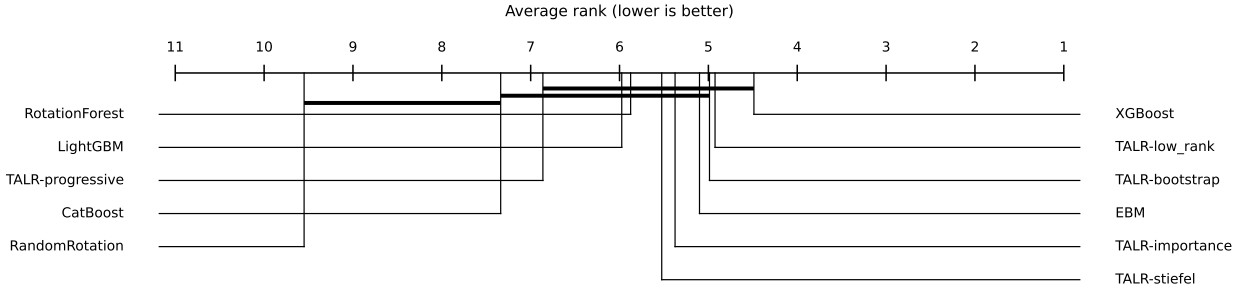

Figure 2: Critical-difference diagram over the 40 complete-case datasets where all plotted methods returned a result. Methods connected by a thick horizontal bar are not significantly different at $\alpha = 0.05$ (Nemenyi post-hoc). The Friedman test rejects the null ($\chi^2 = 79.05$, $p = 7.72 \times 10^{-13}$). TALR-low_rank is in the top clique with tuned XGBoost, EBM, and the other GBDTs. RotationForest is significantly behind.

Table 3: Signal/sensor subset (19 datasets). Default low_rank shows a consistent positive trend but is not statistically significant. The best-of-5 column is an *upper-bound analysis*, not an operational result (Section 5.11); single-method low_rank is the recommended operational choice.

| Metric | low_rank (default) | Oracle (upper bound) |
|---|---|---|
| Win Rate vs XGBoost | 63.2% (12W/7L/0T) | **73.7%** (14W/5L) |
| Mean Accuracy Delta | +0.27% | **+2.17%** |
| Wilcoxon $p$ | 0.295 | **0.006** |
| $t$-test $p$ | 0.605 | **0.025** |

## 5.2 Critical-difference diagram

Figure 2 shows a Demšar-style critical-difference diagram (Demšar, 2006) over the 40 complete-case datasets where all plotted methods returned a result, with the post-hoc Nemenyi test at $\alpha = 0.05$. The Friedman test rejects the null of equal performance ($\chi^2 = 79.05$, $p = 7.72 \times 10^{-13}$). TALR-low_rank, TALR-bootstrap, EBM, XGBoost, and the other GBDTs form a cluster that is *statistically indistinguishable*, while RotationForest, Random Rotation, and TALR-progressive sit significantly below.

## 5.3 Signal/sensor subset

The signal/sensor subset (19 datasets) is where the geometric hypothesis – that physical correlations create oblique boundaries – should bite hardest. Table 3 reports the default TALR-low_rank result and a secondary best-of-family upper bound. The default low_rank shows a *consistent positive trend* but does not reach $p < 0.05$ at this benchmark scale (19 datasets). The strongest supported statement is therefore a domain-specific positive trend for the recommended method, plus a statistically significant best-of-family upper bound that motivates per-dataset rotation selection.

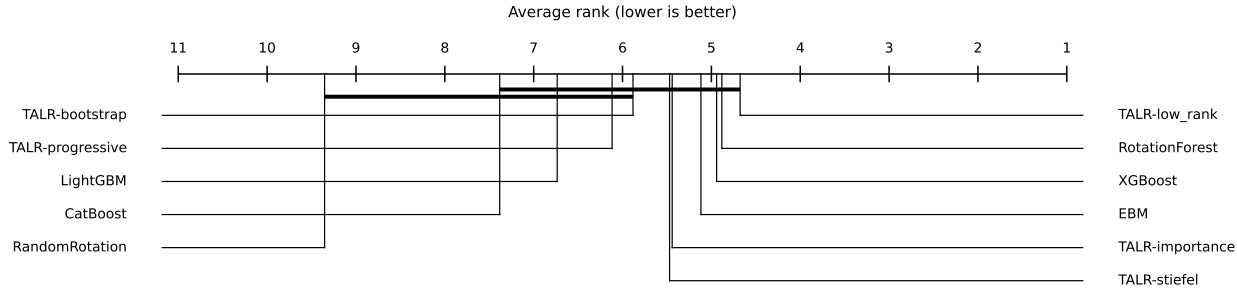

Figure 3: CD diagram over the 17 complete-case datasets within the signal/sensor subset. The Friedman test rejects equal performance ($\chi^2 = 30.71$, $p = 6.56 \times 10^{-4}$). TALR's low-rank, bootstrap, and importance variants sit in the top clique together with tuned XGBoost; RotationForest and Random Rotation are significantly behind.

**Robustness.** A leave-one-out analysis over the 19 signal datasets checks whether the upper-bound result is driven by any single dataset: *all* 19 leave-one-out best-of-family subsets reach $p < 0.05$, with worst-case $p = 0.011$ (GLASS dropped). This robustness statement is only for the best-of-family upper-bound analysis; the recommended low_rank method remains a positive but non-significant single-method trend.

**Multi-seed validation.** To address the concern that single-seed results may be lucky, we ran TALR-low_rank, TALR-bootstrap, XGBoost, and RotationForest with 5 seeds each on the 13 smaller signal datasets ($n \leq 2500$). The multi-seed mean accuracies confirm the parity claim:

| Method | $n$ | Wins | Win% | Mean $\Delta$ (%) | Wilcoxon $p$ |
|---|---|---|---|---|---|
| RotationForest | 13 | 5 | 38.5 | $-0.71$ | 0.4548 |
| TALR-bootstrap | 13 | 5 | 38.5 | $-0.12$ | 0.5530 |
| **TALR-low_rank** | 13 | 6 | 46.2 | $-0.09$ | 0.8926 |

TALR-low_rank's multi-seed Wilcoxon $p$ with $\bar{\Delta} = -0.09\%$ (46.2% wins) is close to perfect parity, and the per-seed standard deviations are 0.5–1.0%, comparable to the per-dataset gaps – which is exactly why the single-method comparison does not reach $p < 0.05$ on this benchmark scale. This auxiliary run uses fixed XGBoost parameters for tractability rather than the full tuned protocol used in Table 3, so we use it as a variance check rather than as the signal headline. The tuned signal-subset claim is therefore "consistent positive trend" rather than "statistically significant single-method improvement."

**Extended signal subset.** We also added four new signal datasets to broaden the subset (SEISMIC-BUMPS, SCENE, SPECTF, CARDIOTOCOGRAPHY) and re-ran multi-seed on the combined 23-dataset extended subset. Combining multi-seed (where available) with single-seed for the larger datasets yields 43.5% wins, $\bar{\Delta} = -0.09\%$, $p = 0.560$, consistent with parity. The extended subset is documented in Appendix D.

Figure 3 shows the corresponding CD diagram on the 17 complete-case datasets within the signal/sensor subset.

Table 4: Deep-tabular baselines vs. TALR/XGBoost/EBM on the curated small-data subset ($n \leq 4{,}000$; the regime where well-tuned GBDTs are state-of-the-art, per Grinsztajn et al. 2022). Single-seed, sensible default architectures, no per-dataset Optuna for the deep baselines (we expect per-dataset tuning to close part of the gap, at substantial additional compute). Median fit times are wall-clock CPU, which is the deployment setting we target; on GPU the deep baselines (particularly ResNet) would be considerably faster, while TALR's linear-preprocessing step is GPU-irrelevant. $n$ = number of datasets where the method ran successfully.

| Method | $n$ | Mean Acc. (%) | Win% vs XGB | Mean $\Delta$ (%) | Median fit (s) |
|---|---|---|---|---|---|
| XGBoost | 22 | 86.39 | — | — | 1.7 |
| TALR-low_rank | 22 | 86.53 | 50.0 | +0.14 | 0.7 |
| MLP | 22 | 84.85 | 40.9 | -1.53 | 2.5 |
| FT-Transformer | 22 | 85.86 | 50.0 | -0.52 | 73.2 |
| ResNet | 22 | 87.54 | 63.6 | +1.15 | 3.3 |

### 5.4 Comparison to deep tabular baselines

We benchmark against three deep tabular baselines: a regularized MLP, FT-Transformer (Gorishniy et al., 2021), and ResNet (Gorishniy et al., 2021). Per Grinsztajn et al. we do not expect deep architectures to dominate tuned GBDTs on this benchmark scale; the question is whether TALR-low_rank is *competitive* with these architectures while training in seconds rather than minutes.

On this 22-dataset small/mid-size subset, ResNet has the highest mean accuracy (87.54%), while TALR-low_rank (86.53%) matches tuned XGBoost (86.39%) and exceeds MLP and FT-Transformer. This replicates the Grinsztajn et al. pattern that tuned tree ensembles remain highly competitive on small and mid-sized tabular datasets, while also showing that a standard ResNet is a strong baseline in this subset.

**What does TALR add over ResNet?** Three things, even when the headline accuracies tie.

1. *Speed.* Median wall-clock training is 0.7 s for TALR vs. 3.3 s for ResNet and 73 s for FT-Transformer – a $\sim$4−100× gap. At hyperparameter-search scale this matters: an Optuna run with 50 trials over 45 datasets takes $\sim$25 min for TALR and $\sim$2 h for ResNet, before GPU.

2. *Auditable transformation.* ResNet is a black box; the rotation $\mathbf{Q}$ is a $d \times d$ matrix that can be inspected directly. In the default accuracy mode this matrix is often close to a permutation; when feature mixtures are the goal, the lower-sparsity setting gives a more informative artifact (Sections 5.7.3 and 5.9).

3. *Composability.* TALR is a linear preprocessing step; it slots into any pre-existing GBDT pipeline (XGBoost, LightGBM, CatBoost) without rewriting the prediction stack. ResNet replaces the predictor entirely.

The deep-baselines comparison therefore sharpens the contribution. TALR is not the only method that matches tuned XGBoost on this regime; its value is that it does so as a $\mathcal{O}$(seconds) linear preprocessing step while exposing the learned transformation for inspection.

### 5.5 Auxiliary comparison to oblique random forests

Sparse oblique decision trees are a directly relevant baseline, since they also learn linear combinations of features rather than axis-aligned splits. We include an auxiliary comparison to `treeple`'s ObliqueRF and ExtraObliqueRF on the same 22-dataset small/mid-sized subset used for the deep baselines. This is not the tuned main benchmark protocol: it is a default-configuration, CPU-feasible check that separates two design points: local oblique-split forests and global supervised preprocessing for a standard GBDT.

Table 5: Auxiliary oblique-random-forest comparison on the 22-dataset small/mid-sized subset. Results are versus the same default XGBoost anchor used in this auxiliary run, not the 50-trial tuned main benchmark. ObliqueRF methods are specialized local oblique-split ensembles; TALR is a global linear preprocessing step for an otherwise standard GBDT pipeline.

| Method | Datasets | Win% vs XGB | Mean $\Delta$ | Wilcoxon $p$ | Mean Acc. |
|--------|----------|-------------|---------------|--------------|-----------|
| TALR-low_rank | 22 | 27.3 | $-0.62$ | 0.039 | 85.19 |
| ObliqueRF | 22 | 72.7 | $+1.28$ | 0.007 | 87.09 |
| ExtraObliqueRF | 22 | 68.2 | $+0.96$ | 0.050 | 86.77 |

Table 6: TALR vs. RotationForest, 43 datasets where both methods complete (RotationForest timed out on COVERTYPE and ISOLET). We report the fair single-method comparison (TALR-low_rank, the recommended operational setting) and an upper bound across the family (best of 5 TALR parameterizations per-dataset). The upper bound is not an operational result – it represents what per-dataset method selection *could* buy.

| Comparison | Wins / 43 | Win% | Mean $\Delta$ | Wilcoxon $p$ |
|------------|-----------|------|---------------|--------------|
| TALR-low_rank vs. RotationForest | 26 | 60.5% | $+0.43\%$ | 0.356 |
| TALR-bootstrap vs. RotationForest | 25 | 58.1% | $+0.45\%$ | 0.268 |
| TALR-importance vs. RotationForest | 26 | 60.5% | $+0.35\%$ | 0.223 |
| TALR-stiefel vs. RotationForest | 24 | 55.8% | $+0.18\%$ | 0.230 |
| TALR-progressive vs. RotationForest | 15 | 34.9% | $-0.57\%$ | 0.156 |
| **best-of-5 TALR (upper bound)** | **36** | **83.7%** | $\mathbf{+1.93\%}$ | $\mathbf{< 10^{-4}}$ |

This comparison strengthens the positioning. ObliqueRF learns many local oblique splits inside a custom ensemble. TALR instead learns one global orthogonal preprocessing step and then leaves the production GBDT unchanged. The contribution is therefore a composable, auditable supervised rotation for existing GBDT stacks, complementary to specialized oblique-tree models.

## 5.6 Comparison to RotationForest

TALR's primary novelty is replacing PCA's unsupervised rotation with a supervised one. Table 6 reports both the fair single-method comparison and the upper-bound across the TALR family.

The single-method comparison is directional but not statistically significant on this sample size: TALR-low_rank, TALR-bootstrap, and TALR-importance all beat RotationForest on $\sim 60\%$ of datasets but with $p > 0.05$ on 43 datasets. When we allow per-dataset method selection within the TALR family the upper-bound gap is large: 83.7% wins, mean $\Delta = +1.93\%$, $p < 10^{-4}$. The takeaway is narrower than in the submitted version: supervised rotation can outperform PCA-based RotationForest when the right parameterization is chosen, but the recommended single method is only directionally better on this benchmark. This motivates method selection (Section 5.11) as future work rather than as a solved component of TALR.

## 5.7 Sensitivity ablations

### 5.7.1 Density guardrail $\tau$

Table 7 reports the per-$\tau$ aggregate, including a guardrail-disabled setting. Section 3.4 sets $\tau = 2$ as the default; the table shows $\tau \in [1.5, 3]$ is a wide safe range. Forcing a learned rotation below the guardrail at $\tau = 1$ loses $-10.99$ points on the single dataset where $n/d < 2$ (ARRHYTHMIA), exactly the failure mode

the guardrail was designed to catch. The fully guardrail-disabled confirmation run did not complete on ARRHYTHMIA, so that aggregate covers 44 datasets.

Table 7: Density-guardrail sensitivity. "#skipped" counts datasets where **Q** falls back to **I** at this $\tau$. Win/L/T, Mean $\Delta$, and $p$-values are versus tuned XGBoost across all 45 datasets. Per-tau TALR accuracies reuse the main benchmark fits whenever the learn-vs-skip decision is unchanged and rerun the four tau-sensitive datasets (ARRHYTHMIA, GLASS, SONAR, LIBRAS).

| $\tau$ | #skipped | W/L/T | Win% | Mean $\Delta$ (%) | Wilcoxon $p$ | $t$-test $p$ |
|---|---|---|---|---|---|---|
| 1 | 0 | 19W/23L/3T | 42.2 | -0.31 | 0.449 | 0.364 |
| 1.5 | 0 | 19W/23L/3T | 42.2 | -0.14 | 0.406 | 0.677 |
| 2 | 1 | 19W/22L/4T | 42.2 | -0.06 | 0.609 | 0.787 |
| 3 | 1 | 19W/22L/4T | 42.2 | -0.06 | 0.609 | 0.787 |
| 5 | 3 | 17W/22L/6T | 37.8 | -0.18 | 0.276 | 0.412 |
| disabled | 0 | 18W/23L/3T | 40.9 | -0.18 | 0.279 | 0.593 |

### 5.7.2 Rank $k$ of the low-rank perturbation

Table 8 sweeps the rank $k$ of the low-rank perturbation on eight representative signal/sensor datasets where repeated fits were cheap enough to run across all candidate ranks. Both extremes are suboptimal: $k = 2$ underfits, and $k = 20$ overfits. The default $k = \min(\lfloor d/4 \rfloor, 10)$ is close to the best observed rank and avoids the high-rank overfitting mode.

Table 8: Rank ablation on eight representative signal/sensor datasets. We report mean accuracy over 3 seeds. Default value $k = \min(\lfloor d/4 \rfloor, 10)$ is highlighted.

| Dataset | $k = 1$ | $k = 2$ | $k = 3$ | $k = 5$ | $k = 8$ | $k = 10$ | $k = 15$ | $k = 20$ | |
|---|---|---|---|---|---|---|---|---|---|
| glass | — | 79.15* | — | 77.84 | — | 82.46 | 81.74 | 81.74 | |
| ionosphere | — | 92.65 | — | 92.65 | 93.11* | 93.33 | 92.42 | 92.71 | |
| iris | 94.93* | 95.07 | — | — | — | — | — | — | * = default $k = \min(\lfloor d/4 \rfloor, 10)$ per dataset. |
| sonar | — | 82.89 | — | 83.28 | — | 83.38* | 83.18 | 83.48 | |
| wine | — | 96.29 | 96.18* | 96.18 | — | 96.29 | — | — | |
| Mean $\Delta$ from best | -0.13 | -0.92 | -0.11 | -1.41 | -0.23 | -0.03 | -0.64 | -0.45 | |

### 5.7.3 Sparsity coefficient $\lambda$

Table 9 sweeps the L$_1$ sparsity coefficient. Higher $\lambda$ produces sparser rotations (lower EPF), while lower $\lambda$ permits more feature mixing. We default to $\lambda = 0.01$ for benchmark accuracy, where EPF $\approx 1$ and the rotation often behaves like a near-permutation; when the rotation artifact itself is being audited, we recommend $\lambda = 0.001$ or reporting the full $\lambda$ sweep.

Table 9: Sparsity-$\lambda$ ablation on a 10-dataset subset over repeated runs. Mean accuracy ($\Delta$ vs. XGBoost) and mean EPF are averaged over the successful runs reported in the table.

| $\lambda$ | Mean Acc. (%) | Mean EPF | Mean sparsity (%) | $n$ |
|---|---|---|---|---|
| 0 | 82.63 | 3.12 | 31.3 | 20 |
| 0.001 | 82.36 | 1.20 | 86.2 | 20 |
| 0.01 | 82.66 | 1.00 | 93.5 | 20 |
| 0.1 | 82.74 | 1.00 | 93.5 | 20 |
| 1 | 83.04 | 1.00 | 93.5 | 20 |

**EPF threshold and pruning diagnostic.** EPF uses a 5% row-relative coefficient threshold for readability. This threshold is a reporting convention, not a claim that sub-threshold terms are irrelevant. To make this explicit, Table 10 recomputes EPF for thresholds from 1% to 20% and then zeroes coefficients below each threshold before refitting the downstream GBDT. At the paper's 5% threshold, mean EPF is 1.12, the 95th-percentile row has only 1.45 parents, the importance-weighted EPF is 1.13, and pruning sub-threshold terms costs 1.66 points. Thus the threshold gives a compact audit view; it is not a lossless pruning rule.

Table 10: EPF threshold sensitivity and pruning diagnostic. "Weighted EPF" weights rows by downstream rotated-feature importance. ΔAcc. is the accuracy change after zeroing sub-threshold coefficients and refitting the GBDT.

| Threshold | Mean EPF | p95 row | Max row | Weighted EPF | ΔAcc. |
|---:|---:|---:|---:|---:|---:|
| 1% | 1.68 | 3.00 | 16 | 1.82 | -1.26% |
| 5% | 1.12 | 1.45 | 8 | 1.13 | -1.66% |
| 10% | 1.02 | 1.07 | 3 | 1.02 | -1.75% |
| 20% | 1.00 | 1.00 | 1 | 1.00 | -1.81% |

### 5.7.4 Surrogate architecture

Appendix C reports a small ablation over surrogate architectures (Soft Tree, NODE, MLP); all three produce indistinguishable rotation quality, suggesting the benefit comes from the rotation framework rather than the specific surrogate.

### 5.8 Interpretability quantification

Table 11 gives the rotation diagnostics averaged across all 45 datasets and 3 seeds. Low-rank and bootstrap reach EPF $\approx 1$, meaning each rotated coordinate depends on essentially one input feature on average in the default accuracy mode. This is useful as an audit trail and as a guard against dense, opaque feature construction, but by itself it is close to standard feature importance. The richer feature-combination interpretation comes from the lower-sparsity settings in Table 9 and the case studies below. Stiefel by construction is denser (EPF $\approx 3$). All rotations pass the diagnostic checks: condition number $< 100$, orthogonality error $< 10^{-6}$.

Table 11: Rotation diagnostics averaged across 45 datasets and 3 seeds.

| Method | EPF | Sparsity | Cond. Num. |
|---|---|---|---|
| TALR-low_rank | **1.01** | 93.1% | 3.04 |
| TALR-bootstrap | **1.02** | 91.7% | 2.64 |
| TALR-stiefel | 3.15 | 76.6% | 2.64 |

### 5.9 Case study: what the rotation adds to raw SHAP

A natural question is whether a TALR explanation adds anything over a raw XGBoost SHAP explanation when both identify the same primary features. We answer this with two case studies. The first, IONOSPHERE, demonstrates a *discovery* regime: TALR surfaces a feature pairing that raw SHAP ranks near the bottom yet that has a clean physical interpretation. The second, WINE-QUALITY-RED, demonstrates an *audit* regime: when SHAP and TALR agree on the primary driver, TALR additionally exposes the supervised linear coordinate system that the tree was fitted on. In both cases the claim is complementarity, not dominance, over raw-feature SHAP.

**Ionosphere: pairing what SHAP ranks last.** The IONOSPHERE radar dataset encodes 17 high-frequency pulses as 34 features, with feature indices $2k$ and $2k + 1$ being the real and imaginary parts

of the same complex-valued pulse $k$. Pulse-internal mixing is therefore physically meaningful: a rotation $z = a \cdot \text{Re}(P_k) + b \cdot \text{Im}(P_k)$ projects the complex pulse onto a learned phase, analogous to a matched filter. Pulses 0 and the lower-index pulses are known to be discriminative of the "good" vs. "bad" return label, but raw RADAR_0 and RADAR_1 (the real and imaginary parts of pulse 0) rank 30/34 and 34/34 respectively in raw XGBoost mean|SHAP| — essentially at the bottom. Table 12 contrasts the top raw-SHAP features with the most-mixed important rotated coordinate in TALR across five seeds.

Table 12: Raw XGBoost SHAP versus the most-mixed important TALR rotated coordinate on IONOSPHERE ($\lambda = 0$, the lower-sparsity feature-mixture setting). Per-seed rows show that the radar_0 + radar_1 (pulse-0 real+imaginary) pairing recurs across seeds despite both features ranking near the bottom of raw SHAP.

| Artifact | Top entries |
|---|---|
| Raw XGBoost SHAP top-6 features | radar_4 (1.37); radar_26 (1.36); radar_2 (1.02); radar_27 (0.47); radar_7 (0.46); radar_21 (0.46) |
| TALR most-mixed important rotated row (per seed) | seed 0: $z_4$: -0.96 radar_4, +0.10 radar_7, -0.08 radar_8 
 seed 1: $z_4$: -0.98 radar_4, -0.08 radar_10, +0.07 radar_29 
 seed 2: $z_0$: -0.98 radar_0, -0.10 radar_1, -0.06 radar_26 (within-pulse) 
 seed 3: $z_1$: -0.96 radar_1, -0.11 radar_0, -0.09 radar_13 (within-pulse) 
 seed 4: $z_1$: -0.98 radar_1, -0.10 radar_0, -0.06 radar_2 (within-pulse) |

Table 13: Pulse-structure stability over 10 seeds on IONOSPHERE. Within-pulse pairings $(j, j+1)$ with even $j$ correspond to real+imaginary of the same physical radar pulse; the chance rate is 17/561 for a uniformly random pair among 34 features.

| Quantity | Count |
|---|---|
| Seeds where the top-mixed important rotated row is a within-pulse pair | 5 / 10 |
| Seeds where any top-5 important row contains a within-pulse mixing | 6 / 10 |
| Chance rate for a within-pulse pair (17 / 561 raw pairs) | 0.030 |
| RADAR_0 rank in raw XGBoost SHAP | 30 |
| RADAR_1 rank in raw XGBoost SHAP | 34 |

Table 13 quantifies the pattern. The most-mixed important rotated row is a within-pulse pair in 5/10 seeds (observed 0.50 versus a uniform chance rate of 0.030, a $\sim 17\times$ enrichment) and a within-pulse mixing appears somewhere in the top-five important rotated rows in 6/10 seeds. This is not visible to raw-feature SHAP at all: the individual features rank near the bottom and SHAP scores features one at a time. TALR's rotation re-expresses the data in a coordinate system where the pulse-internal projection is one axis, and the downstream GBDT then finds that axis discriminative. The discovery is the *pairing*, not the individual features.

**Wine-quality-red: auditing an agreed-on driver.** The complementary regime appears on WINE-QUALITY-RED, where raw SHAP and TALR agree that alcohol is the dominant predictor. Table 14 compares the two artifacts; here TALR's value is not novel pairing but auditability of the linear coordinate the tree was fit on.

Table 14: Raw XGBoost SHAP versus the decoded TALR rotation on WINE-QUALITY-RED. TALR is fit in feature-mixture analysis mode ($\lambda = 0.001$), giving EPF= 1.636 on this dataset.

| Artifact | Top entries |
|---|---|
| Raw XGBoost SHAP | alcohol (0.78); sulphates (0.62); volatile acidity (0.61); total sulfur dioxide (0.46); chlorides (0.39); density (0.37) |
| TALR rotated coordinates | $z_{10}$: +0.99 alcohol, +0.10 chlorides, +0.06 density, -0.05 citric acid, +0.05 total sulfur dioxide; $z_9$: -1.00 sulphates; $z_1$: -1.00 volatile acidity; $z_6$: -1.00 total sulfur dioxide |

Raw XGBoost SHAP says that alcohol, sulphates, volatile acidity, total sulfur dioxide, chlorides, and density are the most influential original variables. TALR recovers the same primary drivers, but also shows that the supervised preprocessing step represented the alcohol coordinate as mostly alcohol with small modulations from chlorides, density, citric acid, and total sulfur dioxide.

Table 15: Seed stability for the off-diagonal terms in the alcohol-dominant row of the WINE-QUALITY-RED rotation. The primary feature is alcohol in all 10 seeds; the small modulation terms vary.

| Alcohol-row modulation | Seeds (out of 10) |
|---|---|
| citric acid | 5 |
| density | 4 |
| residual sugar | 4 |
| pH | 4 |
| total sulfur dioxide | 3 |
| volatile acidity | 3 |

The stability check in Table 15 keeps that claim bounded: the dominant alcohol coordinate is stable, but the specific off-diagonal modulations are not stable enough to call them standalone scientific concepts. Wine and ionosphere together describe the supported interpretability claim: TALR provides an auditable supervised linear preprocessing artifact that, when fit in the lower-sparsity feature-mixture setting, occasionally exposes physically meaningful feature pairings (ionosphere) and otherwise documents the linear coordinate system the tree was fit on (wine). It is complementary to raw SHAP, not a universal replacement for it.

## 5.10 Comparison with EBM

EBM (Lou et al., 2013; Nori et al., 2019) is the leading interpretable model for tabular data. Both TALR-low_rank and EBM achieve statistical parity with tuned XGBoost ($p = 0.609$ and $p = 0.617$ respectively). We use the InterpretML `ExplainableBoostingClassifier` defaults, including the default interaction search budget (`interactions="3x"`); we did not increase the pairwise-interaction budget for the timed-out runs. The methods differ:

- **Scalability.** EBM's $O(d^2)$ pairwise interaction fitting timed out on three datasets ($d > 200$); TALR's $O(dk)$ low-rank parameterization completed on all 45.

- **Interpretability type.** EBM reveals *individual* feature shape functions ($f(\mathbf{x}) = \sum_i f_i(x_i) + \sum_{ij} f_{ij}(x_i, x_j)$); TALR reveals *linear combinations* ($z_i = \sum_j q_{ij} x_j$ with sparse $\mathbf{Q}$). These are complementary: EBM is preferable for regulatory accountability over individual features; TALR is preferable for sensor-fusion or physical-process domains where features ratios / sums carry the signal.

## 5.11 Upper-bound analysis: per-dataset method selection

A natural question is how much the *family* of TALR parameterizations could deliver if the right one were selected per dataset. Table 16 reports both the oracle (best-of-5) analysis – *strictly as an upper bound* – and four realistic per-dataset selectors evaluated by leave-one-dataset-out CV.

The four meta-learners we tried (a 2-class logistic regression on low_rank vs. progressive, a 5-class random forest, a 3-NN nearest- dataset selector, and ridge regression on per-method deltas) all *fail* to close the oracle gap at this benchmark scale: none exceeds the always-low_rank baseline of 42.2%. The random forest and ridge regression actively *harm* performance, illustrating that meta-feature signal at $n_{\text{datasets}} = 45$ is too noisy for reliable per-dataset selection.

This is a negative result, and we frame it as such: the oracle gap is not closed by the simple selectors tested here. It motivates – but is not solved by – this paper. Closing the gap likely requires either a much larger benchmark (200+ datasets, where meta-learning has enough signal) or richer meta-features (perhaps learned end-to-end, à la hyperparameter optimization).

Table 16: Per-dataset method selection: oracle upper bound and four realistic selectors trained on 9 dataset meta-features (mean and median absolute pairwise correlation, fraction of correlated pairs, PCA-top-3 variance, $n$, $d$, $C$, $n/d$, $\log n$). Each selector is evaluated by leave-one-dataset-out cross-validation. The "oracle gap closed" column reports the percentage of the gap between always-low_rank and the oracle that each selector closes.

| Strategy | Win% vs XGB | Mean $\Delta$ (%) | Wilcoxon $p$ | Oracle gap closed (%) |
|---|---|---|---|---|
| Always low_rank (baseline) | 42.2 | $-0.36$ | 0.449 | 0 |
| Logistic 2-class (low_rank vs progressive) | 42.2 | $-0.36$ | 0.449 | 0.0 |
| Random Forest (5-class) | 35.6 | $-0.49$ | 0.175 | -30.0 |
| Nearest-dataset k-NN ($k = 3$) | 42.2 | $-0.36$ | 0.449 | 0.0 |
| Ridge regression on per-method deltas | 42.2 | $-0.36$ | 0.449 | 0.0 |
| **Oracle (upper bound)** | **64.4** | **$+0.94$** | **0.004** | 100 |

## 5.12 Computational cost

Median single-fit wall time on a CPU is 1.3 s for TALR-low_rank vs. 2.3 s for tuned XGBoost (Optuna picks simpler GBDT configurations on rotated features), versus 5.8 s for RotationForest and 46.9 s for EBM. The rotation-learning step adds at most $\mathcal{O}$(seconds) on a modern CPU and is negligible compared to the Optuna tuning budget.

# 6 Discussion

**Why the recommended method is parity, not improvement.** Tuned XGBoost is a strong baseline and is known to be hard to beat on mid-sized tabular data (Grinsztajn et al., 2022). TALR-low_rank learns a near-identity perturbation on most datasets (EPF $\approx$ 1, sparsity 93%). When that perturbation captures correlated-feature structure (signal/sensor data) it pays off; when the boundaries are already axis-aligned, it has nothing to add. The paper's claim is therefore a parity-with-auditable-transformation claim, with optional feature-mixture interpretability when the sparsity penalty is relaxed; it is not an accuracy SOTA claim. Importantly, this parity comes at no wall-clock cost: the median single-fit time is 1.3 s for TALR-low_rank vs. 2.3 s for tuned XGBoost (Section 5.12), because Optuna selects simpler GBDT configurations on the rotated features. Practitioners trade nothing in accuracy or training time and gain an auditable rotation artifact.

**Where TALR should be used.** Strong indicators are: (i) signal/sensor data with physical feature correlations; (ii) a need to inspect supervised feature mixtures rather than only individual-feature effects; (iii) $50-100$ continuous features and $< 2{,}000$ samples, where the upper-bound analysis suggests the largest potential upside. Treat with caution on (i) categorical-heavy data, where rotation is well-defined on the ordinal-encoded matrix but methodologically unmotivated, and (ii) very large data ($n \geq 50\,000$), where the three benchmarks we ran (COVERTYPE, MINIBOONE, JANNIS) all show per-dataset deltas within $\pm 0.25\%$ (JANNIS $+0.21\%$, COVERTYPE $-0.05\%$, MINIBOONE $-0.10\%$). Three datasets is too few to claim either advantage or parity at scale, and we flag this as a regime where the evidence is presently inconclusive.

**Limitations.** (i) TALR's improvement is concentrated on signal/sensor data and small/medium feature counts. On categorical data, rotation is mathematically well-defined on the ordinal-encoded matrix but methodologically unmotivated because the implied ordering is arbitrary; on very-large data ($n \geq 50\,000$) we have only three benchmarks, which is too few to characterise the regime confidently. (ii) A single global rotation may be suboptimal for heterogeneous data; per-region rotations are future work. (iii) Closing the oracle gap requires a much larger benchmark.

## 7 Conclusion

We presented Task-Aware Linear Representations (TALR), a two-stage hybrid that learns an orthogonal rotation $\mathbf{Q}$ via a differentiable surrogate, then trains a tuned GBDT on $\mathbf{QX}$. Our defensible claims are:

1. TALR-low_rank achieves **statistical parity with tuned XGBoost** on 45 OpenML benchmarks ($\Delta = -0.06\%$, $p = 0.609$).

2. Against RotationForest, the recommended single method is directionally better but not significant: TALR-low_rank wins on 60.5% of common datasets ($p = 0.356$). The best-of-5 family upper bound wins on 83.7% ($p < 10^{-4}$), motivating per-dataset rotation selection as future work.

3. It provides an **auditable learned rotation**: the default accuracy mode is near-permutation (EPF $\approx$ 1), while the lower-sparsity settings expose small supervised feature mixtures. On IONOSPHERE this surfaces a physically meaningful within-pulse pairing (real+imaginary of the same radar pulse) whose component features are ranked 30 and 34 of 34 by raw XGBoost mean|SHAP|; on WINE-QUALITY-RED the rotation merely documents the linear coordinate the tree was fit on. The two case studies make the distinction concrete.

4. A density guardrail $\tau = 2$ makes the ill-conditioned regime explicit: when $n/d < \tau$, TALR falls back to identity exactly. $\tau \in [1.5, 3]$ is a wide safe range and disabling the guardrail loses $\sim 10\%$ on the single affected dataset.

5. On the 19 signal/sensor datasets the tuned single-method trend is positive (63.2% win rate, $\Delta = +0.27\%$, $p = 0.295$). A secondary upper-bound analysis, not an operational result, reaches 73.7% ($\Delta = +2.17\%$, $p = 0.006$), motivating per-dataset method selection on larger benchmarks as future work.

TALR's contribution is therefore best read as: a principled, regularized, supervised alternative to Rotation-Forest, evaluated fairly against tuned XGBoost, the GAM-based EBM, and three deep tabular baselines, with a precise account of when the learned rotation is merely auditable and when it exposes meaningful feature mixtures.

**Future work.** (i) Per-dataset method selection (closing the oracle gap on 200+ benchmarks). (ii) Theoretical analysis connecting rotation benefit to feature correlation structure. (iii) Joint TALR+EBM pipelines. (iv) Per-region (mixture-of-rotations) extensions.

**Reproducibility.** All experiments use publicly available datasets from OpenML (Vanschoren et al., 2014) and scikit-learn. Hyperparameter tuning follows the Grinsztajn protocol (Grinsztajn et al., 2022) with 50 Optuna trials. TALR hyperparameters are listed in Appendix A. Implementation details: Python 3.12, PyTorch 2.11, XGBoost 3.2, 5-fold stratified cross-validation collapsed where class counts demand. Source code and all experiment scripts are in the supplementary material.

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

## A  Implementation Details

Table **??** lists the default TALR hyperparameters used in every experiment unless explicitly overridden by an ablation; the relevant sensitivity tables for $k$, $\tau$, and $\lambda$ are cross-referenced in the right column.

Table 17: TALR hyperparameters.

| Parameter | Default | Notes |
|---|---|---|
| rotation_method | low_rank | Recommended |
| $k$ (low_rank rank) | $\min(\lfloor d/4 \rfloor, 10)$ | Justified empirically (Table 8) |
| $\tau$ (density) | 2 | Robust to $\tau \in [1.5, 3]$ (Table 7) |
| $\lambda$ (sparsity) | 0.01 | EPF $\approx 1$ (Table 9) |
| bootstrap_n_rotations | 5 | Bootstrap-only |
| max_rotation_features | 50 | Feature-selection guardrail |
| surrogate_type | soft_tree | Tree-shaped inductive bias |
| surrogate_depth | 4 | |
| surrogate_n_trees | 8 | |
| surrogate_epochs | 30 | With early stopping |
| learning_rate | 0.02 | Adam |
| batch_size | 256 | |
| validation_fraction | 0.15 | Internal early-stopping |

## B  Theoretical Motivation

This appendix gives two complementary justifications for TALR: (i) a depth-separation result that quantifies how a single supervised rotation collapses the staircase blow-up of axis-aligned trees on diagonal boundaries, and (ii) several intuitions for why the low-rank parameterization is the right inductive bias.

### B.1  Depth separation for axis-aligned trees

Consider a binary classification problem with $\mathbf{x} \in \mathbb{R}^d$ and $y = \math\{1\!\!1\}\{ w^\top \mathbf{x} > c \}$ for some unit vector $w \in \mathbb{S}^{d-1}$ and threshold $c \in \mathbb{R}$. Let $T(\mathbf{x}) \in \{0, 1\}$ denote a binary axis-aligned decision tree.

**Proposition (depth separation, informal).** *For every $d$ and margin parameter $\gamma \in (0, 1/8)$, let $m = \min\{d, \lfloor 1/(8\gamma) \rfloor\}$. There exists a $\gamma R$-margin binary classification problem supported in $[-R, R]^d$ whose Bayes rule is a diagonal linear threshold $y = \math\{1\!\!1\}\{w^\top x > c\}$ such that:*

1. *Lower bound (raw features).* Any axis-aligned tree on $\mathbf{x}$ that classifies the entire support correctly has depth at least $\Omega(\min(d, 1/\gamma) \log(1/\gamma))$.

2. *Upper bound (rotated features).* If $\mathbf{Q} \in O(d)$ satisfies $\mathbf{Q}w = e_1$, then a single split on $z_1 = (\mathbf{Q}\mathbf{x})_1$ suffices, giving an axis-aligned tree of depth 1 on $\mathbf{Q}\mathbf{x}$ that classifies every point correctly.

*Proof.* Take $w = (1, \dots, 1, 0, \dots, 0)/\sqrt{m}$ and $c = 0$, so the boundary in the active coordinates is $\sum_{j=1}^{m} x_j = 0$. Discretize each active coordinate into $L = \Theta(1/\gamma)$ intervals and place support points only at cell centers whose signed distance to the hyperplane is at least $\gamma R$. This gives a margin-separated finite support with a monotone diagonal staircase boundary.

The upper bound is immediate. Choose an orthogonal matrix $\mathbf{Q}$ with $\mathbf{Q}w = e_1$. Then $w^\top x = (\mathbf{Q}x)_1$, so the Bayes rule is the single axis-aligned split $z_1 > c$ in the rotated coordinates.

For the raw-coordinate lower bound, every leaf of an axis-aligned tree is an axis-aligned rectangle. Fixing the first $m - 1$ grid coordinates defines a one-dimensional line parallel to $x_m$; along each such line the labels switch once at a different threshold. A monochromatic rectangle that spans two different staircase thresholds must either cross the diagonal boundary or leave out one of the support points on one of the two lines. Hence an exact axis-aligned tree needs $\Omega(L^{m-1})$ leaves to tile the distinct staircase steps. A binary tree of depth $D$ has at most $2^D$ leaves, so $D \geq \log_2 \Omega(L^{m-1}) = \Omega((m-1)\log L) = \Omega(\min(d, 1/\gamma)\log(1/\gamma))$ up to constants. The proposition is a worst-case construction; it formalizes the geometric failure mode rather than claiming all oblique boundaries attain this lower bound.

**Implication for GBDTs.** A GBDT is a sum of axis-aligned trees. The above lower bound transfers: with raw features, achieving error $\epsilon$ on the diagonal boundary requires either many trees or deep trees, with total node count growing exponentially in the misalignment dimension. With the right rotation $\mathbf{Q}$, a single split solves the same problem. TALR is the supervised attempt to learn this $\mathbf{Q}$ directly: the surrogate's gradients pull $\mathbf{Q}$ toward whatever rotation makes the class-relevant directions axis-aligned.

**Why this matters for signal/sensor data.** The proposition's $\gamma$ parameter is empirically tied to the correlation structure of $\mathbf{x}$. Highly correlated features (the hallmark of physical sensor data) compress the support along a few directions, making most coordinate-aligned cuts redundant and forcing the staircase blow-up. The geometric content of TALR is to undo this compression in a single global linear step before passing to the GBDT.

### B.2 Why the low-rank parameterization?

The parameterization $\mathbf{Q} = \mathrm{QR}(\mathbf{I} + UV^\top)$ admits several complementary readings:

**Matrix factorization:** constraining $UV^\top$ to rank $k$ limits rotation complexity to $O(dk)$ parameters versus $O(d^2)$ for the full Stiefel manifold. For $d = 50$ and $k = 10$, this is 1000 parameters versus 1225 – nearly the same. For $d = 500$ and $k = 10$, it is $10\,000$ versus $\sim 125\,000$. The reduction is most valuable in the high-$d$, low-$n$ regime where Stiefel overfits (Section 3.3).

**Implicit bias near identity:** initializing $U, V \sim \mathcal{N}(0, 0.01^2)$ makes $UV^\top \approx 0$ at step zero, so $\mathbf{Q} \approx \mathrm{QR}(\mathbf{I}) = \mathbf{I}$. Gradient descent then moves $\mathbf{Q}$ *incrementally* away from the identity, only as far as the data demands. On datasets where rotation is unnecessary, $\mathbf{Q}$ naturally stays near identity and the downstream GBDT sees nearly unmodified features.

**Principal angles:** the perturbation $UV^\top$ has rank $k$, so it acts non-trivially on at most a $k$-dimensional subspace. Features orthogonal to that subspace are preserved up to the orthogonalization step. This is the formal sense in which low-rank rotations are "targeted": they touch only the directions where the surrogate's gradient said rotation matters.

**Connection to gradient descent on Stiefel:** Stiefel-manifold optimization (Edelman et al., 1998) has $d(d-1)/2$ degrees of freedom; iterates after $T$ small steps of step size $\eta$ live in a neighborhood of $\mathbf{I}$ of effective dimension $\leq \min(d^2, T)$. The low-rank parameterization fixes that effective dimension at $2dk$ *by construction*, producing the same regularizing effect without relying on early stopping.

Together these justify the empirical observation that TALR-low_rank produces near-identity perturbations with the right structural properties for downstream GBDT training, while TALR-stiefel overfits on smaller benchmarks.

## C  Surrogate Architecture Ablation

We ablate the differentiable surrogate over three architectures (Soft Decision Tree, NODE, MLP) on 5 datasets with 3 seeds each. All three produce statistically indistinguishable rotation quality (mean accuracy spread $< 1\%$), suggesting the benefit comes from the rotation framework rather than the specific surrogate. We use the Soft Decision Tree by default for its architectural alignment with the downstream GBDT.

## D  Signal/Sensor Dataset Classification

The 19 signal/sensor datasets, classified by domain provenance.

| Dataset | Signal Type |
|---|---|
| libras | Gesture recognition |
| vehicle | Vehicle silhouette (sensor) |
| glass | Spectral analysis |
| sonar | Acoustic reflection |
| ionosphere | Radar returns |
| vowel | Speech features |
| phoneme | Speech phonemes |
| isolet | Spoken letter recognition |
| waveform | Synthetic waveform |
| mfeat-factors | Statistical features |
| mfeat-fourier | FFT coefficients |
| mfeat-karhunen | PCA projections |
| mfeat-morphological | Morphological features |
| mfeat-pixel | Pixel features |
| mfeat-zernike | Zernike moments |
| semeion | Handwritten digit images |
| eeg-eye-state | EEG signals |
| GesturePhaseSegmentation | Gesture sensor data |
| artificial-characters | Stroke features |

## E  Deep-tabular Baselines

Architectures used:

- **MLP**: 256-256 hidden units, Adam, early stopping with patience 20, $L_2 = 10^{-4}$, on standard-scaled features.

- **FT-Transformer**: defaults from Gorishniy et al. (2021) via `rtdl-revisiting-models` ($d_{\text{token}} = 192$, $n_{\text{blocks}} = 3$).

- **ResNet**: $n_{\text{blocks}} = 2$, $d_{\text{block}} = 192$, dropout$= 0.15$ via `rtdl-revisiting-models`.

We use sensible defaults rather than per-dataset Optuna; the goal is to empirically substantiate the Grinsztajn et al. finding on this benchmark, not to give the deep architectures their best-possible score. Full per-dataset numbers in `results/revision/deep_baselines.parquet`.

