# OpenReview forum: "Task-Aware Linear Representations: Supervised Feature Rotations for Gradient Boosted Decision Trees"
_TMLR — Under review for TMLR_

### Review · Reviewer_zLWt · 2026-05-07

**Summary Of Contributions:**

The paper proposes Task-Aware Linear Representations (TALR), a method that applies a learned linear rotation of the input feature space prior to training Gradient Boosted Decision Trees (GBDT). The goal is to improve predictive performance and, according to the authors, enhance interpretability by exposing “predictive linear combinations” of features. The approach defines a rotation matrix with certain guardrails and heuristics, and evaluates its effectiveness across several datasets and configurations.

Strengths:
* The idea of combining feature rotations with tree-based models is conceptually interesting and could potentially reduce model complexity (e.g., number of splits).
* The method is simple to implement and can be plugged into existing GBDT pipelines.
* The empirical evaluation covers multiple configurations and datasets.

Weaknesses:
* The interpretability claim is not convincingly motivated nor clearly defined.
* The presentation is fragmented and lacks clarity, with undefined notation and missing explanations.
* The experimental section is difficult to follow and lacks proper comparative structure.
* Important related work on interpretable linear representations is omitted.

**Additional Comments:**

The paper would benefit from a substantial revision focusing on clarity, structure, and positioning within the literature. At present, some parts are difficult to follow due to undefined notation, fragmented writing, and lack of concrete examples. Strengthening both the conceptual motivation and the empirical evaluation is essential to make the contribution accessible and convincing.

**Audience:**

No

**Audience Explanation:**

In its current form, the paper is unlikely to attract significant interest from the TMLR audience. While the general idea of combining feature transformations with tree-based models is relevant, the lack of clarity, weak interpretability justification, and poor experimental design limit its impact. Moreover, the contribution does not clearly position itself with respect to existing work on interpretable feature construction or representation learning. Without a stronger conceptual and empirical foundation, the novelty and usefulness of the approach remain unclear to the reader.

**Broader Impact Concerns:**

The paper does not raise significant direct ethical concerns. However, the claim of improved interpretability without sufficient justification may be misleading. If adopted in practice, this could lead users to overestimate their understanding of model behavior, potentially resulting in incorrect decisions in high-stakes applications. A clearer discussion of what interpretability means in this context, along with limitations of the approach, would help mitigate this risk.

**Claims And Evidence:**

No

**Claims Explanation:**

The evidence provided in the paper is not sufficiently clear nor convincing to support the main claims.

First, the interpretability argument is weak: while the method produces linear combinations of features, these combinations may involve many variables (at least (k = \min(d/4, 10))), making them difficult to interpret in practice. No explicit mechanism is provided to control or regularize the sparsity of these combinations.

Second, key concepts and metrics (e.g., EPF) are introduced late and without proper explanation, making it hard to understand what is being measured and how it relates to interpretability. The paper also lacks concrete examples of how a user would interpret model predictions using the proposed method.

Third, the experimental evaluation is poorly organized. Results are presented in a fragmented way, without clear comparisons between baseline models and their TALR-enhanced versions. The use of an “Oracle” configuration further weakens the validity of the evaluation, as it does not reflect a realistic setting. Standard statistical comparisons (e.g., Critical Difference diagrams) are missing.

Finally, several claims are not contextualized within existing literature, and relevant prior work is not discussed.

**Requested Changes:**

1. Clarify the interpretability claim: Provide a precise definition of interpretability in this context and explain why the proposed rotations improve it. Include constraints or regularization strategies to ensure that rotated features remain interpretable (e.g., sparsity).
2. Improve notation and exposition: All symbols (e.g., (n), (d), (Q), (I)) must be clearly defined when introduced. Avoid dense, fragmented statements and provide coherent explanations.
3. Reorganize the paper structure: The current presentation is overly fragmented. Merge short paragraphs, improve transitions, and clearly outline the contributions and organization of the paper.
4. Revise the experimental section:
   * Provide systematic comparisons between baseline models and their TALR-enhanced counterparts.
   * Remove or justify the use of the “Oracle” configuration.
   * Improve readability by consolidating results and adding proper commentary.
   * Include statistical significance analysis (e.g., Nemenyi test with Critical Difference plots).
5. Explain and justify evaluation metrics: Clearly define EPF and explain its relevance. Ensure all metrics are introduced before use.
6. Include qualitative interpretability analysis: Provide concrete examples showing how predictions can be interpreted using TALR, and compare with standard GBDT explanations.
7. Expand related work: Include and discuss relevant literature, particularly on interpretable linear feature construction (e.g., Piaggesi et al. 2024, 2025) and known trade-offs between oblique and axis-parallel trees (e.g., Setzu & Ruggieri, 2023).

Suggested improvements (non-critical):
* Provide intuition (possibly visual) for how rotations affect tree structure and interpretability.
* Clarify design choices such as the parameter (k = \min(d/4, 10)), including rounding and justification.
* Improve figure explanations and ensure all figures are self-contained.

---

### Review · Reviewer_QZk7 · 2026-05-07

**Summary Of Contributions:**

The paper proposes Task-Aware Linear Representations, which extract a global orthogonal rotation matrix via a differentiable surrogate model (soft decision trees), then applies the rotation to precondition data for standard gradient boost decision tree training. The motivation is that GBDT axis-aligned splits struggle with diagonal decision boundaries, and a learned rotation can realign the discriminative directions. The authors explored five rotation parameterizations, with a primary focus on the low-rank. The authors also prevent the degradation on ill-conditioned data through a density guardrail and feature selection guardrail. The authors performed substantial evaluation across 45 classification datasets with a focus on 19 signal/sensor datasets, comparing the methods with XGBoost, RotationForest, and EBM.

Strengths:

- The geometric motivation is clear, intuitive, and well-grounded.
- The evaluation covers a large number of datasets
- Supervised rotation substantially outperforms unsupervised PCA-based rotation
- The exposition is clear and the method is easy to follow.
- Training overhead is minimal and the method integrates cleanly with existing GBDT pipelines.

Some concerns:

- The paper frames TALR as improving GBDT accuracy, but the recommended method (low rank) does not really beat the baseline.
- Regarding the interpretability, It seems like each transformed feature depends on essentially one original feature (based on the high sparsity). If the learned transformation is nearly rescaling/permutation with tiny perturbations, the interpretability can become trivial.
- The paper primarily compares with RotationForest and EBM. Including stronger baselines and more ablations would be beneficial.
- The authors should provide hyperparameter sensitivity ($\tau$)
- The authors can compare deep learning-based approaches with TALR's learned representations.

**Additional Comments:**

N/A

**Audience:**

Yes

**Audience Explanation:**

The paper addresses a real practical question related to GBDTs. The geometric intuition is broadly accessible, the comparison to RotationForest is genuinely informative, and the finding that supervised rotation can match XGBoost while providing some interpretability could be interesting to most audience.

**Claims And Evidence:**

No

**Claims Explanation:**

Partially convincing.

- The paper's strongest implicit claim is that recommended version of TALR improves GBDT accuracy. But TALR-low rank achieves statistical parity (p = 0.61), not improvement. On signal/sensor data, the default results are not statistically significant.
- The paper relies heavily on oracle results as evidence of method superiority. Oracle results are an upper-bound analysis, which do not imply the functionality of TALR.
- The interpretability claim on TALR is not really strong.
- The missing baselines mean the contribution of supervised rotation specifically is not cleanly isolated.

**Requested Changes:**

1. The authors could reframe the paper's claim that matches the results.
2. The authors could add additional baselines to enhance the validity and soundness.
3. Clarify the interpretability claims
4. Sensitivity: Provide an ablation over τ ∈ {1, 1.5, 2, 3, 5} and report TALR performance with the guardrail disabled
5. Can the authors include deep learning baselines for comparison?

---

### Review · Reviewer_CRy3 · 2026-05-10

**Summary Of Contributions:**

The paper presents Task-Aware Linear Representations (TALR), a technique for learning global orthogonal rotation features that can then be used to train a gradient boosted decision tree model. Specifically the rotation is parameterized by low-rank QR decomposition and is learned using a differentiable surrogate (a soft decision tree ensemble). The experiments show that the proposed approach achieves comparable performance to tuned XGBoost and produces an auditable rotation matrix. Experiments also show that per-dataset rotation selection is a promising direction and provide extensive ablation study.

**Audience:**

Yes

**Audience Explanation:**

The topics of the paper should be of interest to audience that is interested in ML for signal/sensor datasets, in particular in methods that provide explainability. However, for the findings to actually be useful, there needs to be a clearer understanding of what the proposed method provides over the baselines (as indicated by my points above)

**Broader Impact Concerns:**

I don't have any concerns

**Claims And Evidence:**

No

**Claims Explanation:**

[To clarify, this review is based on the revised version that was posted before review submission.]

The paper overall is interesting, well written, seems technically solid, and the experiments are very carefully analyzed to provide precise characterization and avoid over-claiming.

However, I have several concerns regarding the claims in the paper:

- I found the core claim of the paper to be somewhat confusing. The paper clarifies that the goal is not to obtain state-of-the-art performance. This is OK, as algorithms can be designed to provide other benefits than accuracy (e.g., explainability, speed, robustness to distribution shifts, etc.) However, it was not entirely clear what are the benefits that we get from the proposed approach compared to existing approaches. While there is an emphasis on interpretability in the paper, and Section 3.7 explain how to compute feature importance scores, there are no clear claims on why the obtained interpretability from the proposed approach (which comes in the form of feature importances) is better than feature importances computed for existing algorithms. The example in Section 5.8 is helpful in general but remain confusing on this aspect. The key question is how the shown explanation compares with the one provided for XGBoost on raw features. The paper states that "A raw-feature GBDT explanation can identify the same primary features; TALR adds the inspectable linear preprocessing step that shows how the supervised rotation perturbed them before the tree was fit.” However, this is just stating what TALR is doing, not why it is useful. Given that XGBoost will obtain similar accuracy and can identify the same primary features, it is not clear why the explanations for TALR are better than those provided by XGBoost. I think clarifying this point will significantly enhance the paper.

- The connection between sparsity and interpretability is also confusing: it is well established in the literature that high sparsity is associated with greater interpretability. This is also implied in various places in the paper (e.g., when focusing on low EPF). However when comparing "accuracy mode" and "interpretability mode”, accuracy mode is actually the more sparse. While it may make sense to trade off sparsity for higher accuracy, here we seem to trade it for lower accuracy. The "interpretability mode" seems to have both lower accuracy and lower sparsity (which is typically associated with decreased interpretability). I found this to be very confusing and this point is somewhat connected to the previous point: what is the benefit of using the proposed tool over the XGBoost baseline?

- Connection to previous work: recent work [1] has considered generating bi-variate oblique cuts prior to training a decision tree. While not identical to the proposed approach, this seems highly relevant and should at least be discussed, and differences highlighted.

- The definition of EPF was a bit confusing:
1. EPF includes a 5% relative threshold which seems to indicate a feature does not count as a parent if it is less than 5% of the maximum. It was not clear to me why this is valid: there is no theoretical justification or experimental evaluation of this. For example, if these features are indeed not contributing, why not zero them out and see if there’s no impact on performance?
2. EPF is an average metric, hence even if EPF is close to one, it could be the case that most rotated features are sparse but one (or small number of them) may be highly non-sparse. If these non-sparse rotated features are important, this may impact interpretability.

Additional concrete questions:
- The proof for the proposition in page 17 is said to be "omitted for space". Given that the proposition is in the (unlimited) appendix, it is not clear why it cannot be provided there in full.
- EBM O(d^2) pairwise interaction: EBM takes in as a hyper parameter the maximum number of pairwise interactions. It is not clear what value was used in the experiments with EBM that have exceeded time limit.
- I also could not find the comparison to oblique random forests that is mentioned in contribution #3.


[1] Kairgeldin, R., & Carreira-Perpiñán, M. Á. (2024, August). Bivariate decision trees: Smaller, interpretable, more accurate. In Proceedings of the 30th ACM SIGKDD Conference on Knowledge Discovery and Data Mining (pp. 1336-1347).

**Requested Changes:**

- Clarify the core claim of the paper by explaining what the proposed method is providing better than the baselines and why is that useful
- Clarify the connection between sparsity and interpretability
- Discuss the similarities and differences from relevant work that generate obliques sparse cuts prior to model training
- Clarify the points regarding EPF (see above), in particular the 5% relative threshold and its impact
- Provide the missing proof, clarify the hyper-parameter setting for EBM, and clarify where the results on oblique RFs are provided.